# Health conditions associated with overweight in climacteric women

**Maria Suzana Marques**[1,2], **Ronilson Ferreira Freitas**[1], **Daniela Araújo Veloso Popoff**[1,2], **Fernanda Piana Santos Lima de Oliveira**[2]*, **Maria Helena Rodrigues Moreira**[3], **Andreia Maria Araújo Drummond**[4‡], **Dorothéa Schmidt França**[2‡], **Luís Antônio Nogueira dos Santos**[1,2‡], **Marcelo Eustáquio de Siqueira e Rocha**[1‡], **João Pedro Brant Rocha**[4‡], **Maria Clara Brant Rocha**[5‡], **Maria Fernanda Santos Figueiredo Brito**[1], **Antônio Prates Caldeira**[1,2], **Fabiana Aparecida Maria Borborema**[2‡], **Viviane Maria Santos**[2‡], **Josiane Santos Brant Rocha**[1,2]

1 State University of Montes Claros, Montes Claros, Minas Gerais, Brazil, 2 Fipmoc University Center (UNIFIPMoc), Montes Claros, Minas Gerais, Brazil, 3 University of Trás-dos-Montes and Alto Douro, Department of Sports Science, Exercise and Health, Vila Real, Portugal, 4 Federal University of Minas Gerais, Belo Horizonte, Minas Gerais, Brazil, 5 Faculty of Medical Sciences of Minas Gerais, Belo Horizonte, Minas Gerais, Brazil

☯ These authors contributed equally to this work.
‡ These authors also contributed equally to this work.
* fernandapiana@gmail.com

**Data Availability Statement:** All relevant data are within the paper and its Supporting Information files.

## Abstract

This study aims to investigate the association between health conditions and overweight in climacteric women assisted by primary care professionals. It is a cross-sectional study conducted with 874 women from 40 to 65 years of age, selected by probabilistic sampling between August 2014 and August 2015. In addition to the outcome variable, overweight and obesity, other variables such as sociodemographic, reproductive, clinical, eating and behavioural factors were evaluated. Descriptive analyses of the variables investigated were performed to determine their frequency distributions. Then, bivariate analyses were performed through Poisson regression. For the multivariate analyses, hierarchical Poisson regression was used to identify factors associated with overweight and obesity in the climacteric period. The prevalence of overweight and obesity was 74%. Attending public school (PR: 1.30–95% CI 1.14–1.50), less schooling (PR: 1.11–95% CI 1.01–1.23), gout (PR: 1.18–95% CI 1.16–1.44), kidney disease (PR: 1.18–95% CI 1.05–1.32), metabolic syndrome (MS) (PR: 1.19–95% CI 1.05–1.34) and fat intake (PR: 1.12–95% CI 1.02–1.23) were considered risk factors for overweight. Having the first birth after 18 years of age (PR: 0.89–95% CI 0.82 to 0.97) was shown to be a protective factor for overweight and obesity. The presence of overweight and obesity is associated with sociodemographic, reproductive, clinical and eating habits.

**Funding:** The authors received no specific funding for this work.

**Competing interests:** The authors have declared that no competing interests exist.

## Introduction

Brazil has been experiencing a rapid process of demographic and epidemiological transition, leading to the frequent occurrence of chronic degenerative diseases[1]. The increase in the prevalence of overweight, represented by overweight and obesity, among the elderly female population raises great concern in developed and developing countries. Since overweight and obesity are risk factors for adverse health events[2], such as disturbances in lipid and glucidic metabolism, psychological stress and sleep alterations, there is an increased risk of cardiovascular diseases[3], musculoskeletal disease, acute myocardial infarction[4], cancer[5] and worse quality of life[6] among patients who are overweight and obese in comparison to those who are satisfied with their body weight[7].

Overweight and obesity have become public health problems worldwide. The projection for 2025 is that approximately 2.3 billion adults will be overweight, and more than 700 million will be obese. According to a study conducted in 2016, the rate of overweight among Brazilian women is 50.5%, and this this frequency increases with age up to 64 years[8].

Epidemiological data associating excess weight with behavioural and clinical variables in climacteric women[9], using probabilistic samples[10], are still scarce. Considering that the climacteric period is an important part of the life cycle of women and that this period is related to the potential peak of fat mass and obesity in this group, the current study aimed to investigate the association between health conditions and an excess weight ratio in climacteric women assisted by primary care professionals, since this phase may assume pathological characteristics or be associated with other chronic diseases.

## Materials and methods

This is a component study of the project entitled "Health problems of climacteric women: an epidemiological study", conducted in the city of Montes Claros, Minas Gerais, Brazil, whose central theme is the health of climacteric women. This project was developed by a group of researchers and considers the central theme in the following lines of research: metabolic syndrome, mental health, obesity, quality of life, sleep disorders, health perception, urinary incontinence, perception of climacteric symptoms and levels of physical activity; each of these themes was developed by researchers who make up the research group.

A cross-sectional and analytical study was carried out in the city of Montes Claros, Minas Gerais, Brazil, from August 2014 to August 2015; the target population consisted of 30,801 climacteric women enrolled in 73 health care units, excluding pregnant, postpartum or bedridden women. This study was carried out in the Family Health Strategy (FHS) system, which represents the primary health care (PHC) mechanism in the public health system in Brazil[11].

Sampling was of the probabilistic type, and the selection of the sample occurred in two stages. Each health care unit team was taken as a conglomerate, from which 20 units were drawn, covering the urban and rural areas for data collection. Following this stage, a proportional number of women were randomly selected according to the climacteric stratification criteria of the Brazilian Society of Climacteric women (SOBRAC), in 2013[12]. For each unit, 48 women were selected; a total of 960 women summoned. To incorporate the structure of the complex sampling plan in the statistical analysis of the data, each interviewee was associated with a weight (w), which corresponded to the inverse of their probability of inclusion in the sample (f)[13]. Women between 40 and 65 years of age who were enrolled in the selected teams and physically able to respond to the questionnaires and be submitted to anthropometric measurements and laboratory tests (12-hour fasting) were considered eligible to participate in the study. The researchers previously trained all data collectors and interviewers and maintained supervision during the data collection stage. After training the interviewers and prior to

the actual data collection, a pilot study was conducted in a unit of the FHS, with women belonging to the age group studied and not part of the final sample. The pilot study allowed the questionnaire and the interviewers' performance to be tested in practice. After this phase, the field research was started. Adjustments to the data collection instrument were not required. After selection, the women were invited to arrive for research participation on a previously established date. The final sample consisted of 874 climacteric women who were invited to sign the informed and post-informed consent forms.

Overweight and obesity, which was considered the outcome variable of this work was evaluated by body mass index (BMI). Despite the inclusion of some patients who were over 60 years old, women were categorized into eutrophic (BMI <25 kg/m$^2$) and overweight (IMC $\geq$ 25 kg/m$^2$), following a categorization model used in other studies with similar population groups[14, 15, 16]. Initially, women were weighed wearing light clothing and without footwear, in an orthostatic position, with their feet together and arms relaxed beside the body, by a mechanical anthropometric medical scale (Balmak 11[R]) with a capacity of 150 kg and weight increments divided into 100g. The stature was measured by an anthropometer (SECA 206[R]) that was fixed to a flat wall and was without skirting. In this measurement, the women were instructed to keep their feet together and stand in an upright position, with their head positioned in the Frankfurt plane. For the calculation of BMI, the body weight in kilograms was divided by the squared height, expressed in metres (BMI = P/A$^2$).

The women answered questions related to the independent variables, which were allocated in three blocks: (1) sociodemographic, (2) reproductive, and (3) clinical, eating and behavioural factors.

The block of sociodemographic variables included age (40–45, 46–51, 52–65 years); type of school (public, private); level of schooling (elementary school I, elementary school II, high school or higher education); marital status (married, separated, divorced, widowed); labour occupation (yes, no); monthly income ($\geq$ 01 minimum wage, <01 minimum wage), where the minimum wage was equivalent to US $217,42 at the time of data collection; number of people residing in the same house (up to 2, more than 2); and skin colour (white, not white).

The reproductive variables comprised the age of menarche ($\leq$ 11 years, 12–14 years and $\geq$ 15 years), first birth weight (<4000 g; $\geq$ 4000 g), climacteric symptoms assessed by the Kupperman index[17] (absent/mild; moderate/severe) and age at first delivery ($\leq$ 18 years old, > 18 years).

The clinical, eating and behavioural variables included liver disease (absent, present), gout (absent, present), renal disease (absent, present), metabolic syndrome (MS) (absent, present); urinary incontinence (absent, present), cardiovascular disease risk (low risk, intermediate risk, high risk), drinking (yes, no), fat intake (yes, no), smoking (yes, no), symptoms of depression, quality of sleep and physical activity.

Metabolic syndrome (MS) was evaluated using the Third Report of the National Cholesterol Education Program Expert Panel on Detection, Evaluation, and Treatment of High Blood Cholesterol in Adults (NCEP-ATPIII) criteria of the Brazilian Society of Diagnosis and Treatment of MS[18]; urinary incontinence was assessed by the International Consultation on Incontinence Questionnaire-Short Form ICIQ-SF[19]; the risk for cardiovascular diseases was assessed by the Framingham Global Risk Score[20]; the symptoms of depression were evaluated by the Beck Depression Inventory[21]; sleep quality was assessed by the Pittsburgh Sleep Quality Index[22]; and physical activity practice was assessed through the International Physical Activity Questionnaire (IPAQ short version)[23].

The women were submitted to peripheral venous blood collection to analyse the laboratory parameters. Serum triglyceride levels were determined by the colourimetric enzymatic method. The level of high-density lipoprotein (HDL) cholesterol was obtained by selective

precipitation of ((low-density lipoprotein (LDL) cholesterol and very low-density lipoprotein (VLDL) cholesterol with dextran sulfate in the presence of magnesium ions, followed by dosing by the enzymatic system cholesterol oxidase/peroxidase with calorimetry and reading, as performed in the total cholesterol dosage, using Labtest®️ reagents, in a Cobas Mira®️[24] apparatus. The lipid profile was analysed according to parameters proposed by the Brazilian Society of Cardiology[25] and fasting glycaemia according to the standards of the Expert Committee on the Diagnosis and Classification of Diabetes Mellitus[26].

The data were tabulated in the statistical software Statistical Package for Social Science (SPSS, version 21, Chicago, Illinois). Initially, descriptive analyses of all variables were carried out to determine their frequency distributions, and then, bivariate analyses of the outcome variable with each independent variable were performed using the chi-square test. Gross prevalence ratios (PRs) were estimated with their respective 95% confidence intervals. Variables with a descriptive level (p-value) of less than 0.25 were selected for multivariate analysis using the hierarchical Poisson regression model, adapted to the model proposed by other authors [10]. The model was composed of blocks of distal (sociodemographic variables), intermediate (reproductive) and proximal (clinical, eating and behavioural) variables. Adjusted prevalence ratios (PRs) with their respective 95% confidence intervals were estimated, and only those that presented a descriptive level of $p < 0.05$ remained in the model. At each hierarchical level, the stepwise forward procedure was adopted: the statistically significant variables selected in the bivariate analysis started in the model, and then other variables were added (Fig 1).

As this study involved humans, it was submitted, evaluated and approved for execution by the Research Ethics Committee of the Faculdades Integradas Pitágoras (Protocol: 817.666).

## Results

The sample consisted of 874 women between 40 and 65 years of age, of whom 74.1% were overweight and obese. When categorized by climacteric status, it was observed that postmenopausal women had a higher prevalence of overweight/obesity (54.3%).

The results of the bivariate analysis revealed that the following variables were associated with the overweight and obesity outcome: age between 52 and 65 years (p = 0.184), private school attendance (p = 0.000), less schooling (p = 0.093) (p = 0.0006), liver disease (p = 0.000), gout (p = 0.000), kidney disease (p = 0.106), weight of the 1st child at birth equal to or greater than 4000 g (p = .039), high risk for cardiovascular diseases (p = 0.000), alcohol consumption (p = 0.039) and fat intake (p = 0.065). However, women between 46 and 51 years of age (p = 0.184), who had a late menarche age (p = 0.039) and had children over 18 years old (p = 0.004) experienced a protective effect against overweight and obesity. It should be emphasized that there was a high prevalence of overweight and obesity in all the independent variables presented (Table 1).

Some sociodemographic (marital status, monthly income, number of individuals residing in the same house and colour of skin), clinical and behavioural (smoking, physical activity, depression symptoms, sleep quality) factors did not present significant associations (p <0.250) with overweight and obesity and were not included in the hierarchical model.

The health conditions that were associated with overweight and obesity in the hierarchical model at the distal level were private school attendance (PR = 1.30, p = 0.000) and low level of education (PR = 1.11, p = 0.033). After adjusting for sociodemographic factors, an association at an intermediate level between age at first childbirth above 18 years (PR = 0.90, p = 0.010) was observed, and this variable had a protective effect against the occurrence of overweight and obesity (Table 2). At the proximal level, after adjusting for the potential confounding factors analysed, the presence of gout (RP = 1.18, p = 0.004), MS (PR = 1.29, p = 0.000), kidney

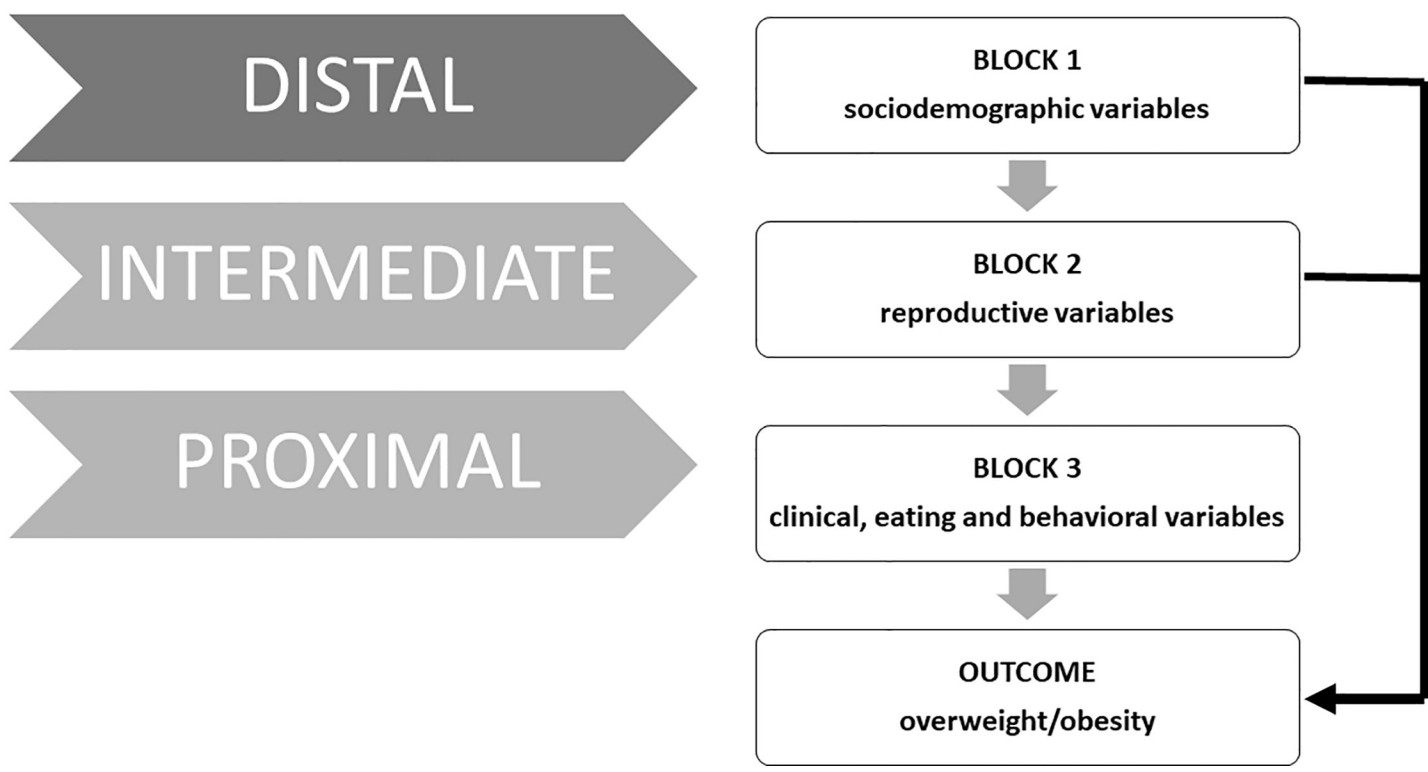

**Fig 1. Model with the statistically significant variable selected in the bivariate analysis, and then adding other variables.**

disease (PR = 1.19, p = 0.006) and fat intake (PR = 1.12, p = 0.014) were found to be positively associated with overweight and obesity (Table 2).

## Discussion

The prevalence of overweight and obesity in the population of the present study was higher than 2/3 of the sample, with a mean BMI of 28.67 ± 6.35 kg/m$^2$ and with a predominance of overweight in postmenopausal women. These findings are in accordance with a study conducted in São Paulo/Brazil, where the mean BMI in postmenopausal women was 29.0 ± 5.6 kg/m2[27].

Weight gain in climacteric women is due to the ageing process and oestrogenic depletion, with a centralized distribution of fat mass related to ovarian failure[28], which leads to a change in the hormonal environment previously dominated by oestrogen to an environment where there is a predominance of testosterone, favouring androgenicity[29]. Additionally, inadequate lifestyle habits, such as a sedentary lifestyle and the consumption of fats and sugars, can lead to physiological and metabolic alterations[30]. The limited perception of body weight and the importance of its control[31] and the use of medications such as antidepressants, analgesics, and anxiolytics[32] also compete for a role in this condition.

Obesity is associated with insulin resistance and chronic inflammation predisposing obese individuals to various diseases, including breast cancer, whose pathogenesis has been linked to increased oestrogen levels[33].

In addition, excessive body weight also contributes to the occurrence of systemic arterial hypertension (SAH), depression and worsening of climacteric symptoms[34]. Together with

**Table 1. Sample characteristics and gross prevalence ratios (PRs) for overweight and obesity women according to the sociodemographic, reproductive, clinical, behavioural and eating factors of menopausal women.**

| Variables | | n | %* | Overweight/obesity (%)* | Gross PR (CI$_{95\%}$) | p-value |
|---|---|---|---|---|---|---|
| **Sociodemographic** | | | | | | |
| Age | 40 to 45 | 236 | 27.9 | 73.2 | 1.00 | 0.184 |
| | 46 to 51 | 241 | 26.8 | 70.0 | 0.95 (0.85–1.07) | |
| | 52 to 65 | 397 | 45.4 | 77.0 | 1.04 (0.95–1.15) | |
| Type of school attended | Public | 822 | 97.3 | 73.2 | 1.00 | 0.000 |
| | Private | 24 | 2.7 | 93.6 | 1.26 (1.11–1.43) | |
| Schooling | High school/Graduate | 281 | 31.8 | 70.9 | 1.00 | 0.093 |
| | Fundamental II | 231 | 26.6 | 73.0 | 1.03 (0.92–1.15) | |
| | Fundamental I | 358 | 41.6 | 77.5 | 1.11 (1.01–1.21) | |
| Labour occupation | Yes | 347 | 40.4 | 71.7 | 1.00 | 0.106 |
| | No | 520 | 59.6 | 76.0 | 1.07 (0.99–1.16) | |
| **Reproductive** | | | | | | |
| Age at menarche | 12 to 14 (Normal) | 513 | 60.6 | 75.9 | 1.00 | 0.039 |
| | ≤ 11 (Early) | 101 | 11.8 | 79.8 | 1.06 (0.95–1.18) | |
| | ≥ 15 (Late) | 260 | 27.6 | 67.6 | 0.90 (0.82–1.00) | |
| Weight of 1st child at birth | < 4000 g | 600 | 84.8 | 73.0 | 1,00 | 0.050 |
| | ≥ 4000 g | 106 | 15.2 | 80.8 | 1.11 (1.00–1.24) | |
| Climacteric symptoms | Absent/Light | 541 | 62.3 | 72.6 | 1,00 | 0.203 |
| | Moderate/Intense | 332 | 37.7 | 76.4 | 1.05 (0.97–1.14) | |
| Age at first delivery | ≤18 years | 218 | 27.3 | 81.2 | 1,00 | 0.004 |
| | > 18 years | 605 | 72.7 | 72.1 | 0.89 (0.82–0.96) | |
| **Clinical, eating and behavioural factors** | | | | | | |
| Liver disease | Absent | 792 | 91,6 | 73.0 | 1.00 | 0.000 |
| | Present | 74 | 8.4 | 86.3 | 1.21 (1.10–1.33) | |
| Gout | Absent | 822 | 95.4 | 73.0 | 1.00 | 0.000 |
| | Present | 38 | 4.6 | 91.9 | 1.27 (1.15–1.40) | |
| Kidney disease | Absent | 700 | 85.4 | 72.1 | 1.00 | 0.000 |
| | Present | 119 | 14.6 | 88.2 | 1.20 (1.10–1.31) | |
| Metabolic syndrome | Present | 317 | 35.2 | 59.6 | 1.00 | 0.000 |
| | Absent | 557 | 64.8 | 81.9 | 1.39 (1.25–1.53) | |
| Urinary incontinence | Absent | 676 | 77.5 | 71.9 | 1.00 | 0.026 |
| | Present | 195 | 22.5 | 81.2 | 1.10 (1.01–1.20) | |
| Cardiovascular disease | Low risk | 388 | 43.7 | 66.6 | 1,00 | 0.000 |
| | Intermediate risk | 423 | 48.4 | 78.7 | 1.15 (1.06–1.26) | |
| | High risk | 66 | 7.9 | 87.0 | 1.31 (1.16–1.46) | |
| Alcoholism | No | 646 | 78.8 | 73.0 | 1,00 | 0.239 |
| | Yes | 163 | 21.2 | 79.8 | 1.06 (0.96–1.16) | |
| Fat intake | No | 655 | 80.2 | 73.0 | 1.00 | 0.065 |
| | Yes | 163 | 19.8 | 79.8 | 1.09 (1.00–1.19) | |

* values corrected by the drawing effect (deff); PR: Gross prevalence ratio; 95% CI: Confidence interval.

other comorbidities, excessive body weight impairs the quality of life of women and impacts their functionality[6,35,36].

According to the findings of this study, having attended private school seems to be associated with overweight in the climacteric women. This may be due to an increased accessibility

**Table 2. Adjusted prevalence ratios for overweight and obesity according to sociodemographic, reproductive, clinical, eating and behavioural factors of climacteric women.**

| Variables | | PR (CI$_{95\%}$) adjusted | *p* value |
|---|---|---|---|
| **Sociodemographic (distal level)** | | | |
| Type of school attended | Public | 1.00 | |
| | Private | 1.30 (1.14–1.50) | 0.000 |
| Schooling | High School/Graduate | 1.00 | |
| | Fundamental II | 1.05 (0.94–1.17) | 0.420 |
| | Fundamental I | 1.11 (1.01–1.23) | 0.033 |
| **Reproductive (Intermediate level)** | | | |
| Age at first delivery | ≤18 years | 1.00 | |
| | > 18 years | 0.90 (0.82–0.97) | 0.010 |
| **Clinical, eating and behavioural factors (proximal level)** | | | |
| Gout | Absent | 1.00 | |
| | Present | 1.18 (1.05–1.32) | 0.004 |
| Metabolic syndrome | Absent | 1.00 | |
| | Present | 1.29 (1.16–1.44) | 0.000 |
| Kidney disease | Absent | 1.00 | |
| | Present | 1.18 (1.08–1.29) | 0.000 |
| Cardiovascular disease | Low risk | 1.00 | |
| | Intermediate risk | 1.05 (0.95–1.15) | 0.332 |
| | High risk | 1.19 (1.05–1.34) | 0.006 |
| Fat intake | No | 1.00 | |
| | Yes | 1.12 (1.02–1.23) | 0.014 |

PR: adjusted prevalence ratio; 95% CI: confidence interval

of high caloric foods in childhood and adolescence or maternal obesity during pregnancy[36] that leads to weight excess, which could be perpetuated in adult life. However, the literature cannot explain these findings consistently, presenting evidence of a higher prevalence of weight excess among students of private schools in other age groups[37,38].

Nevertheless, some studies have shown an association between less schooling and high BMI [39], in agreement with the present findings, suggesting that a higher level of education may favour healthier living habits, such as the intake of vegetables and fruits[40] and the regular practice of physical activity[41]. Physical activity, including strength and endurance training, has a significant effect on aspects related to women's health in menopause, including favourable aspects of mineral metabolism, such as iron[42], which may also be influenced by probiotic supplementation, which improves the quality of the impaired intestinal microbiota in obese patients[43].

Regarding the gynaecological aspects, having a first delivery that occurred after the age of 18 was shown to be a protective factor for overweight and obesity. Other studies have also shown an association between overweight and obesity, early parturition and parity[44,45]. Findings suggest that younger maternal age at first delivery is independently associated with a higher risk of central obesity and MS in climacteric women[46]. One explanation would be the possibility of a higher number of pregnancies among women with early parturition and lifestyle changes, although the pathophysiology of this association is still unclear and deserves additional study[47]. Multiparity is associated with an increase in the prevalence of MS since it favours abdominal obesity[48] and insulin resistance in climacteric women[49].

The diagnosis of gout is also associated with overweight and obesity in climacteric women. This finding becomes relevant since hyperuricaemia is correlated with insulin resistance, hypertension, obstructive sleep apnoea, chronic renal disease (CKD), MS and elevated cardiovascular risk[50,51]. According to this context, hyperuricaemia may be related to an increase in the prevalence of coronary artery disease (CAD) and to the incidence of major cardiovascular events in climacteric women as an independent risk factor[52]. Chromosomal abnormalities are associated with elevated serum levels of uric acid and gout in postmenopausal women, demonstrating a possible role of sex hormones in the regulation of the urate transporter in gout[53].

An association between kidney disease and overweight and obesity was found in the present study. These data are consistent with the Brazilian Society of Nephrology's Dialysis Survey in 2014, which showed that 37% of dialysis patients were overweight or obese and that overweight and obesity was as a risk factor for CKD[54]. In addition, obesity was associated with MS, which is also a risk factor for the development of CKD[55].

Overweight is related to compensatory hyperfiltration, which occurs to meet the metabolic demands increased by body weight, with possible damage to the kidneys and increased risk of long-term glomerulopathy, in addition to being a risk factor for nephrolithiasis and kidney cancer. The obese patient also has a higher relative risk for developing albuminuria and a decrease in the glomerular filtration rate, even without CKD[56].

In climacteric women, with increased risk for obesity, MS becomes more prevalent, increasing the incidence of cardiovascular disease and the risk of acute myocardial infarction (AMI) [57], a vulnerability attributed to the decrease of oestrogen and insulin resistance[58]. The association between overweight and obesity and MS was observed in the present study with a consequent risk elevation for cardiovascular diseases. Another study corroborated these findings and demonstrated that the prevalence of MS was also higher in postmenopausal women [59]. Obesity presents as a possible primary factor for the occurrence of MS and the risk of cardiovascular diseases, since an overweight patient may also have visceral adiposity, which is one of the diagnostic criteria of MS.

Among the overweight and obese women in this study, a diet characterized by fat intake was associated with overweight. A document published by the Health Surveillance Agency points out that excessive consumption of saturated fat, as well as sugars, is related to the development of chronic noncommunicable diseases, including obesity[60]. A balance in fat consumption is a viable strategy for a possible reduction of cardiovascular risk in this population [61], since inadequate diet is the leading cause of cardiovascular mortality[25].

The present study presents as limiting factors the use of BMI as the sole diagnostic criterion for overweight and obesity, as opposed to using other gold standard techniques of body analysis, such as Dual X-ray Densitometry (DEXA). The liver diseases, kidney disease and gout variables were measured by self-report, and it was not possible to establish with precision the different aetiologies of these diseases; however, being able to establish their association in a generic way provoked the need for further studies using more accurate diagnostic tools, such as imaging or laboratory tests. Moreover, this was a cross-sectional study and, therefore, it was unable to establish causality among the studied variables. Despite the presented limitations, the study was carried out with methodological rigor, and the obtained results provide relevant information on the subject in addition to listing variables to be studied in future studies. It should be emphasized that the sample used in the study was representative of the population and was obtained in a probabilistic way, strengthening the results and associations obtained.

In addition, from a socioeconomic point of view, the population studied resides in a region that represents the Brazilian reality with confidence; it is located in a transition zone between what is considered rich Brazil (represented by the southern and southeastern states) and

regions of Brazil with characteristics of poverty (represented by the northern and northeastern states). Therefore, the present study reports associations relevant to the health of climacteric women in an emblematic and representative segment of the Brazilian population. These results can be used to implement public policies to assist climacteric women in preventing the occurrence of overweight and its consequences.

## Conclusion

The presence of overweight and obesity was associated with climacteric women who had attended private schools, who had low schooling, gout, metabolic syndrome, and kidney disease, who had high cardiovascular risk and who ingested fats in their diet. In turn, having a first delivery after 18 years of age was presented as a protective factor for women not becoming overweight and obese. Monitoring of these modifiable factors is suggested since they were associated with overweight in climacteric women assisted by primary health care services.

## Supporting information

**S1 File. DATABASE.**
(SAV)

## Author Contributions

**Conceptualization:** Maria Suzana Marques, Ronilson Ferreira Freitas, Daniela Araújo Veloso Popoff, Fernanda Piana Santos Lima de Oliveira, Maria Helena Rodrigues Moreira, Andreia Maria Araújo Drummond, Dorothéa Schmidt França, Luís Antônio Nogueira dos Santos, Marcelo Eustáquio de Siqueira e Rocha, João Pedro Brant Rocha, Maria Clara Brant Rocha, Maria Fernanda Santos Figueiredo Brito, Antônio Prates Caldeira, Fabiana Aparecida Maria Borborema, Viviane Maria Santos, Josiane Santos Brant Rocha.

**Data curation:** Maria Suzana Marques, Ronilson Ferreira Freitas, Daniela Araújo Veloso Popoff, Josiane Santos Brant Rocha.

**Formal analysis:** Maria Suzana Marques, Ronilson Ferreira Freitas, Daniela Araújo Veloso Popoff, Fernanda Piana Santos Lima de Oliveira, Maria Helena Rodrigues Moreira, Andreia Maria Araújo Drummond, Dorothéa Schmidt França, Luís Antônio Nogueira dos Santos, Marcelo Eustáquio de Siqueira e Rocha, Maria Fernanda Santos Figueiredo Brito, Antônio Prates Caldeira, Josiane Santos Brant Rocha.

**Investigation:** Maria Suzana Marques, Ronilson Ferreira Freitas, João Pedro Brant Rocha, Maria Clara Brant Rocha.

**Methodology:** Maria Suzana Marques, Ronilson Ferreira Freitas, Daniela Araújo Veloso Popoff, Josiane Santos Brant Rocha.

**Project administration:** Josiane Santos Brant Rocha.

**Resources:** Josiane Santos Brant Rocha.

**Software:** Josiane Santos Brant Rocha.

**Supervision:** Daniela Araújo Veloso Popoff, Fernanda Piana Santos Lima de Oliveira, Maria Helena Rodrigues Moreira, Andreia Maria Araújo Drummond, Dorothéa Schmidt França, Luís Antônio Nogueira dos Santos, Marcelo Eustáquio de Siqueira e Rocha, Maria Fernanda Santos Figueiredo Brito, Antônio Prates Caldeira, Josiane Santos Brant Rocha.

**Validation:** Josiane Santos Brant Rocha.

**Visualization:** Daniela Araújo Veloso Popoff, Fernanda Piana Santos Lima de Oliveira, Maria Helena Rodrigues Moreira, Maria Fernanda Santos Figueiredo Brito, Antônio Prates Caldeira, Josiane Santos Brant Rocha.

**Writing – original draft:** Maria Suzana Marques, Ronilson Ferreira Freitas, Daniela Araújo Veloso Popoff, Fernanda Piana Santos Lima de Oliveira, Maria Helena Rodrigues Moreira, Marcelo Eustáquio de Siqueira e Rocha, João Pedro Brant Rocha, Maria Clara Brant Rocha, Maria Fernanda Santos Figueiredo Brito, Antônio Prates Caldeira, Josiane Santos Brant Rocha.

**Writing – review & editing:** Maria Suzana Marques, Daniela Araújo Veloso Popoff, Fernanda Piana Santos Lima de Oliveira, Maria Helena Rodrigues Moreira, Andreia Maria Araújo Drummond, Dorothéa Schmidt França, Luís Antônio Nogueira dos Santos, Marcelo Eustáquio de Siqueira e Rocha, Maria Fernanda Santos Figueiredo Brito, Antônio Prates Caldeira, Fabiana Aparecida Maria Borborema, Viviane Maria Santos, Josiane Santos Brant Rocha.

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
