## [Decision Letter · Decision Letter 0]

7 Aug 2019

PONE-D-19-15143

HEALTH CONDITIONS ASSOCIATED WITH OVERWEIGHT IN CLIMACTERIC WOMEN

PLOS ONE

Dear  Authors

Thank you for submitting your manuscript to PLOS ONE. After careful consideration, we feel that it has merit but does not fully meet PLOS ONE’s publication criteria as it currently stands. Therefore, we invite you to submit a revised version of the manuscript that addresses the points raised during the review process. 

Please incorporate the reviewers comments step by step of both the reviewers 

We would appreciate receiving your revised manuscript by Sep 21 2019 11:59PM. To enhance the reproducibility of your results, we recommend that if applicable you deposit your laboratory protocols in protocols.io, where a protocol can be assigned its own identifier (DOI) such that it can be cited independently in the future. For instructions see: http://journals.plos.org/plosone/s/submission-guidelines#loc-laboratory-protocols

We look forward to receiving your revised manuscript.

Kind regards,

Nayanatara Arun Kumar

Academic Editor

PLOS ONE

Journal Requirements:

3. Please provide the full name of the ethics committee which approved this study in the ethics statement in your manuscript and the online submission form.

4.  Please cite and discuss the following related work:  

https://journals.plos.org/plosone/article?id=10.1371/journal.pone.0211617

Please cite and incorporate your response related to this related paper to the body of the manuscript (introduction and discussion). Please bear in mind that our publication criteria state " If a submitted study replicates or is very similar to previous work, authors must provide a sound scientific rationale for the submitted work and clearly reference and discuss the existing literature. Submissions that replicate or are derivative of existing work will likely be rejected if authors do not provide adequate justification.

5. Please remove your figures from within your manuscript file, leaving only the individual TIFF/EPS image files, uploaded separately.  These will be automatically included in the reviewers’ PDF.

6. ** Please include your tables as part of your main manuscript and remove the individual files **. Please note that supplementary tables (should remain/ be uploaded) as separate "supporting information" files.

Reviewers' comments:

Reviewer's Responses to Questions

**Comments to the Author**

1. Is the manuscript technically sound, and do the data support the conclusions?

Reviewer #1: Partly

Reviewer #2: Yes

2. Has the statistical analysis been performed appropriately and rigorously? 

Reviewer #1: Yes

Reviewer #2: Yes

3. Have the authors made all data underlying the findings in their manuscript fully available?

Reviewer #1: No

Reviewer #2: Yes

4. Is the manuscript presented in an intelligible fashion and written in standard English?

Reviewer #1: No

Reviewer #2: Yes

5. Review Comments to the Author

Reviewer #1: Thanks for your invitation. I reviewed the manuscript “PONE-D-19-15143 “(HEALTH CONDITIONS ASSOCIATED WITH OVERWEIGHT IN CLIMACTERIC WOMEN), the following is my comments:

1. It is necessary that for authors to find someone who is familiar with English to revise this paper.

2. The authors seemed to be unclear about the independent variables and outcome variables; we can find this confused expression in both of abstracts and the main paper.

3. Regression models are common used in study these years, authors do need to explain their analysis step by step, and the tables are substandard.

4. Some part of discussion is irrelevant with the results.

In view of the above reasons, I cannot give my specific suggestion about this paper.

Reviewer #2: I was honored to review the manuscript entitled “Health conditions associated with overweight in climacteric women” submitted to Plos One.

The paper presents interesting and good quality study; the aim of this study was to investigate the association between health conditions and overweight in climacteric women assisted by primary care professionals..

I recommend to accept the manuscript after minor revision.

There are also some other points to correct:

- please provide the list of abbreviations.

- did you perform body composition analisis?

- Introduction and Discussion section needs improvement- please cite: doi:

10.1039/C9FO01006H ; 10.20452/pamw.4426. ; 10.1097/MD.0000000000014909.

- In discussion please provide “study strong points” and “study limitation” section.

I recommend to accept the manuscript after minor revision.

6. PLOS authors have the option to publish the peer review history of their article (what does this mean?). If published, this will include your full peer review and any attached files.

Reviewer #1: No

Reviewer #2: No

---

## [Author Response · Author response to Decision Letter 0]

20 Oct 2019

REBUTTAL LETTER

Journal Requirements

RESPONSE: Thank you for your observation. We have reviewed and sought to meet PLOS ONE's style requirements.

RESPONSE: We apologize for the English language errors. This issue has been solved since all the text was submitted for editing by the AJE team.

3. Please provide the full name of the ethics committee which approved this study in the ethics statement in your manuscript and the online submission form.

RESPONSE: Thank you for your observation. The full name of the ethics committee that approved this study has been provided in the ethics statement in our manuscript (Line 181-182).

4. Please cite and discuss the following related work: https://journals.plos.org/plosone/article?id=10.1371/journal.pone.0211617

RESPONSE: Thank you for your observation. The work was cited (Line 58).

The present work is not a replication of prior research but research that is part of a project called “Health problems of climacteric women: an epidemiological study”, which was developed by a research group linked to a postgraduate programme. Strictly speaking, the central theme is the health of climacteric women. This group consists of several researchers who each dealing with one of the following lines of research: metabolic syndrome, mental health, obesity, quality of life, sleep disorders, health perception, urinary incontinence, climacteric symptoms, and levels of physical activity. Thus, the work now presented is a product of the obesity in climacteric women line of research (Line 75-82).

5. Please remove your figures from within your manuscript file, leaving only the individual TIFF/EPS image files, uploaded separately. These will be automatically included in the reviewers’ PDF.

RESPONSE: Thank you for your observation. The figure have been removed from the manuscript file.

6. ** Please include your tables as part of your main manuscript and remove the individual files **. Please note that supplementary tables (should remain/ be uploaded) as separate "supporting information" files

RESPONSE: Thank you for your observation. The tables have been included as part of the main manuscript and removed from their individual files.

Reviewer #1

Comment 1

“It is necessary that for authors to find someone who is familiar with English to revise this paper”.

RESPONSE: We apologize for the English language errors. This issue has been solved since all the text was submitted for editing by the AJE team.

Comment 2

“The authors seemed to be unclear about the independent variables and outcome variables; we can find this confused expression in both of abstracts and the main paper.

RESPONSE: The suggestion was followed, and the text was changed. The outcome variable is overweight/obesity (Line 35/110-111). Thank you for the suggestion.

Comment 3

“Regression models are common used in study these years, authors do need to explain their analysis step by step, and the tables are substandard.”

RESPONSE: The work was performed following the methodological rigor for random sampling, and a considerable number of participants were included. The data collectors were calibrated, and prior to collection, a pilot study was performed. The statistical analysis was conducted appropriately, and the tables were checked by the authors. Thank you for the suggestion (Line 94-97/102-107/ 311-313).

Comment 4

“Some part of discussion is irrelevant with the results”:

RESPONSE: The discussion was reviewed by the authors, including suggestions made by the reviewers. Thank you for the suggestion (Line 249-250/252-254/261-265).

Reviewer #2

Comment 1

“please provide the list of abbreviations.

RESPONSE: The suggestion was followed, and the text was changed. All the meanings of the acronyms have been contemplated in the text. Thank you for the suggestion.

AMI acute myocardial infarction

BDI Beck Depression Inventory

BMI body mass index

CAD coronary artery disease

CI confidence interval

CKD chronic renal disease

DEXA dual X-ray absorptiometry

FHS Family Health Strategy

FSH follicle stimulating hormone

HDL high-density lipoprotein

ICIQ-SF™ International Consultation on Incontinence Questionnaire - Short Form 

IPAQ International Physical Activity Questionnaire

LDL low-density lipoprotein

MG Minas Gerais 

MS metabolic syndrome

NCEP/ATP-III Third Report of the National Cholesterol Education Program Expert Panel on Detection, Evaluation, and Treatment of High Blood Cholesterol in Adults

NHA National Health Agency

NHC National Health Council

PHC Primary Health Care

PR prevalence ratios

S stature

SAH systemic arterial hypertension

SOBRAC Brazilian Climacteric Society

SPSS Statistical Package for the Social Sciences 

VLDL very low-density lipoprotein 

W weight

Comment 2

“did you perform body composition analisis?”

RESPONSE: The study used BMI as a diagnostic criterion of overweight/obesity as opposed to using other gold standard techniques, such as X-ray double emission densitometry (DEXA), which allows the analysis of body composition. However, it should be noted that although the use of BMI is a limiting factor, its use is supported by the acceptance of BMI as a diagnostic criterion for obesity by the World Health Organization (Line 311-323).

Comment 3

 “Introduction and Discussion section needs improvement- please cite: doi: 10.1039/C9FO01006H; 10.20452/pamw.4426.; 10.1097/MD.0000000000014909.”

RESPONSE: These have been cited. Thank you for the suggestion (Line 249-250/252-254/ 261-265).

Comment 4

“In discussion please provide “study strong points” and “study limitation” section.

RESPONSE: This was a randomized, probabilistic sample of a representative size, with methodological rigor, data collector training and a pilot study. All of the study procedures were conducted according to current legislation in relation to research ethics. Study limitation: This was a prevalence study that does not allow the establishment of cause-effect conclusions, and there was no use of DEXA for body analysis (Line 311-323/ 330-332).

---

## [Editor Report · Decision Letter 1]

20 Nov 2019

HEALTH CONDITIONS ASSOCIATED WITH OVERWEIGHT IN CLIMACTERIC WOMEN

PONE-D-19-15143R1

Dear Dr Fernanda Piana Santos Lima de Oliveira

We are pleased to inform you that your manuscript has been judged scientifically suitable for publication and will be formally accepted for publication once it complies with all outstanding technical requirements.

With kind regards,

Nayanatara Arun Kumar

Academic Editor

PLOS ONE

Additional Editor Comments (optional):

Dear authors

I appreciate the efforts and the corrections done by as per the guidance of all the respectable reviewers
---

## [Editor Report · Acceptance letter]

5 Dec 2019

PONE-D-19-15143R1 

Health conditions associated with overweight in climacteric women 

Dear Dr. Piana Santos Lima de Oliveira:

I am pleased to inform you that your manuscript has been deemed suitable for publication in PLOS ONE. Congratulations! Your manuscript is now with our production department. 

With kind regards,

on behalf of

Dr. Nayanatara Arun Kumar 

Academic Editor

PLOS ONE